# Engineering *N*-acyl-homoserine lactone-based quorum-sensing circuit for dynamic regulatory control in *Saccharomyces cerevisiae*
Aafke C. A. van Aalst [1] ✉, Maxence Holtz [1], Mikkel Lyskjær Jensen [1], Tabea Schröder[1], Christoph Crocoll [2], Michal Poborsky [2], Emil Damgaard Jensen [1] ✉ & Michael Krogh Jensen [1]

The yeast *Saccharomyces cerevisiae* is widely employed in industrial biotechnology for chemical and pharmaceutical production. However, engineering yeast for high product titre remains challenging due to metabolic imbalances and competition for cellular resources. To address this, we developed an orthogonal quorum-sensing (QS) system based on *N*-acyl-homoserine lactones (AHLs) for cell density-dependent regulation in yeast. Using metabolic engineering, we established AHL production in yeast. Next, we improved AHL-biosensors via directed evolution and a novel growth-based screening strategy with *amdS* as a counter-selectable marker. We identified three LuxR variants with enhanced sensitivity and confirmed N86 to play an important role in their sensitivity to ligands, corroborating literature on the native system in bacteria. These sensitive LuxR variants were engineered for QS-controlled expression of a reporter gene, demonstrated by delayed autonomous expression of yeGFP. Additionally, we engineered LuxR to function as a repressor, achieving QS-dependent repression. The QS system was applied to enhance aloesone production, a plant-derived metabolite with cosmetic and pharmaceutical applications. The established system showed 51% increased production through QS-controlled repression of *FAS1*. This work establishes a versatile QS-based regulatory platform to support dynamic pathway regulation for metabolic engineering in yeast.

Nowadays, microorganisms such as the yeast *Saccharomyces cerevisiae* can be readily adapted to synthesize products of interest[1]. Engineering yeast for product formation typically involves the introduction of non-native biosynthetic pathways alongside rewiring or deletion of endogenous metabolic pathways[1,2]. However, optimizing these engineered strains for high product titre remains a significant challenge due to metabolic imbalances and increased energy expenditure associated with product biosynthesis. Specifically, anabolic product pathways compete with native metabolism for essential cellular resources, including carbon, energy, and cofactors. Consequently, engineered strains often exhibit suboptimal product titre and reduced growth rates, conferring an evolutionary disadvantage relative to non-engineered strains which can result in genetically unstable yeast strains that lose productivity over time[3]. Developing generic strategies for dynamic regulation of cellular resources to balance growth and product formation is therefore a critical challenge in microbial biotechnology[4].

Transcription factors are natural proteins that have evolved to regulate gene expression in response to key intracellular signals or environmental changes, and have been applied for engineered dynamic pathway regulation[4,5]. The first synthetic network constructed in yeast, based on the bacterial Tet Repressor, enabled timed induction of gene expression by addition of tetracycline or derivatives[6]. Since then, systems for inducible gene expression based on nutrient composition[7] temperature[8] or light[9,10] have been reported. Studies using these inducers showed that delayed expression of genes encoding enzymes of biosynthetic pathways could improve product titre[11,12]. However, inducers are often costly and/or require process-specific medium or cultivation conditions that, for economic, technological and/or safety reasons, can be unsuitable for large-scale fermentations. These complications contribute to a growing interest in inducer-free systems such as two-stage processes linking the expression of genes encoding enzymes of biosynthetic pathways to cell-population density[13].

[1]The Novo Nordisk Foundation Center for Biosustainability, Technical University of Denmark, 2800 Lyngby, Denmark. [2]Department of Plant and Environmental Sciences, University of Copenhagen, 1870 Frederiksberg, Denmark. ✉e-mail: acava@dtu.dk; emdaje@biosustain.dtu.dk

https://doi.org/10.1038/s42003-025-09163-9                                                                                                          **Article**

In nature, microbial populations can regulate gene expression in response to cell density through quorum sensing (QS), a cell-cell communication mechanism mediated by small signalling molecules known as autoinducers (AIs)[14,15]. In Gram-negative bacteria, the most common AIs are N-acyl-homoserine lactones (AHLs), which consist of a homoserine lactone ring attached to an acyl chain of varying length and oxidation state[16,17]. AHLs are synthesized by LuxI-family synthases using S-adenosylmethionine and acyl carrier protein (ACP)-bound acyl groups as substrates[16]. In general, each AHL synthase predominantly synthesizes a single type of AHL and can produce additional AHLs in smaller amounts[18]. The acyl group preference is defined by the substrate specificity of the enzyme rather than by the supply of acyl substrates available in the cytoplasm. Once synthesized, AHLs diffuse into the extracellular environment, where they accumulate in a density-dependent manner. Upon reaching a critical concentration, AHLs can diffuse into the cells and bind to LuxR-type transcriptional regulators, which undergo a conformational change. The regulator dimerizes and binds to a regulator-specific operator sequence (e.g., luxO) within target promoters (Fig. 1A), thereby modulating the expression of various genes involved in processes such as bioluminescence[19], pathogenesis[20] and biofilm formation[21], illustrating the powerful role of QS in coordinating population-wide behaviours.

AHL-based QS systems have been extensively adapted for synthetic biology applications, primarily in bacterial systems[13,14,22–24]. However, their implementation in eukaryotic hosts such as S. cerevisiae remains under-explored. To date, only a limited number of QS-based circuits have been successfully engineered in yeast, including a plant hormone-responsive system[25] and a rewired pheromone signalling pathway[26–28]. However, since these systems consist of eukaryotic elements, their implementation is inherently linked to off-target effects such as growth arrest or morphological changes, challenging their utility in biotechnological applications. Prokaryotic QS systems offer a potentially orthogonal regulatory strategy that minimizes crosstalk with native eukaryotic signalling networks. Moreover, the biochemical diversity of AHL-based QS systems enables the construction of multiple, independently regulated circuits[14], providing a versatile framework for dynamic control of metabolic pathways in yeast.

In this study, we aimed to develop a fully orthogonal, AHL-based QS system for autonomous and tunable regulation of metabolic pathways in S. cerevisiae. To achieve this, we here present the rational and evolution-guided engineering of AHL sensing and a first demonstration of AHL production in yeast. Based on the engineered production and sensing systems, we demonstrate dynamic regulation of a reporter gene, as well as improved aloesone production using the engineered QS-based regulatory platform, thus demonstrating a novel broadly utilizable tool for yeast metabolic engineering.

## Results

### Establishing acylated homoserine lactone production in yeast

In many proteobacteria, acyl-HSLs are produced by synthases from the LuxI protein family. From this family, we selected LuxI (from *Vibrio fischeri*[24]) LasI (from *Pseudomonas aeruginosa*[29]) and EsaI (from *Pantoea stewartia*[29]) as these are well-characterized and are already successfully used in synthetic QS-circuits in bacterial sytems[13,23,24,29]. LuxI and EsaI predominantly catalyze the formation of 3-oxo-C6-HSL, while 3-oxo-C12-HSL is the predominant product of LasI[24,29]. We expressed them in S. cerevisiae (native and codon-optimized version). When genomically integrating a single copy of each of these enzymes driven by the strong Sc.TDH3 promoter in yeast (strains AAA021; luxI, AAA022; lasI, AAA023; esaI, AAA079; luxI_co, AAA081; lasI_co, AAA083; esaI_co), we were not able to detect any acyl-HSL-compounds in the supernatant of the yeast cultures. This indicated that these enzymes were not expressed or directly functionally transferable between bacteria and yeast. In addition to these 3 synthases, we decided to express a codon-optimized version of cepI (Burkholderia cenocepacia) as well, which, in contrast to the previously tested synthases, catalyzes the formation of the straight-chain acyl-HSL C8-HSL. When integrating the expression cassette of Bc.cepI in yeast (strain AAA063), production of C8-

HSL was observed at 7 nM (Figs. 1C, S1). As the substrates of LuxI-type synthases are S-adenosyl-L-methionine (SAM) and an acyl-donor (Fig. 1B), we tried to further increase the production by boosting the precursor supply of SAM, using methionine feeding as well as overexpression of S. cerevisiae's SAM2 and MET6[30]. For the upregulation of SAM2 and MET6, an expression cassette with weak constitutive promoters pACT1 and pPGI1 (indicated by +) as well as an expression cassette with strong constitutive promoters pTEF1 and pPGK1 (indicated by ++) were introduced into yeast (strains AAA111; CepI (++) and AAA113, CepI (+)). Sole introduction of cepI did not affect the growth performance (Figure S2B), consistent with previous findings showing that S. cerevisiae exhibits no phenotypic changes when exposed to AHLs at concentrations up to 200 nM[31].and in agreement with our observation when culturing yeast in the presence and absence of C8-HSL (Fig. S2C). However, in line with literature[30], the introduction of an additional copy of SAM2 and MET6, resulted in a reduction in growth rate of approximately 11% and 20%, for the weak and strong constitutive promoters, respectively, and a 19% reduction in final optical density ($OD_{660}$) for strong constitutive promoters, under microplate conditions (Fig. S2D). Using strong constitutive promoters in addition to methionine feeding resulted in 3.4-fold increase ($26 \pm 2$ nM) in C8-HSL production (Fig. 1C). To rule out the possibility that low titre resulted from degradation of AHLs by S. cerevisiae, we monitored extracellular C8-HSL levels over time in stationary phase cultures. These remained stable (Fig. S2A), indicating that the limiting factor is synthesis efficiency rather than turnover. To minimize any additional fitness trade-offs, we chose not to pursue further improvements of C8-HSL production.

### Acylated homoserine lactone sensing in yeast via LuxR

In order to establish a heterologous QS-system based on acyl-HSL, we next focused on establishing an acyl-HSL biosensor. Recently, Tominaga et al. (2021) have successfully implemented LuxR as a transcriptional activator upon addition of 3-oxo-C6-HSL in S. cerevisiae[32]. However, this system has not been tested for C8-HSL. LuxR-type regulators are known to bind different acyl-HSLs with different specificities[17], and therefore, this system[32] was used as a starting point. As a prototypic design based on the work by Tominaga et al. (2021), we first expressed a variant LuxR with two mutations S116Y and W201R, described to be essential for functionality in yeast[32], fused with a VP48 activation domain and a nuclear localisation signal (NLS) or without these domains. These reading frames were integrated into the genome, together with the GAL core-promoter (pGAL_core) equipped with 5 LuxR-specific operator sequences (luxO) driving the expression of yeGFP (yeast strains AAA019; pGAL_core-5xluxO-yeGFP, AAA036; pGAL_core-5xluxO-yeGFP + VP48-NLS-luxR, AAA035; pGAL_core-5xluxO-yeGFP + luxR). The VP48-NLS-LuxR design showed induction of yeGFP expression by 3-oxo-C6-HSL (Fig. 1D), while the LuxR construct lacking VP48 and NLS showed only moderate repression in response to this ligand (Fig. S3). In its native context, LuxR usually activates gene expression by recruiting the RNA polymerase to the promoter, but can also repress expression by either preventing RNA polymerase binding or blocking its progression along the promoter, depending on the location of the operator sequence[33,34]. LuxR-type regulators recognize operator sequences through a set of essential base-contacting residues, while tolerating variation at non-critical positions[24]. To quickly test whether LuxR could function as a repressor in yeast, we repurposed a pTEF1 promoter variant already available in our lab. This promoter contained a luxo-type operator sequence (similar but not identical to the luxO used in the pGAL_core–5xluxO design) positioned downstream of the TATA-box, as is common when engineering biosensors using prokaryotic repressors[35]. This design was integrated into the genome together with both LuxR variants (yeast strains AAA014; pTEF_luxO*-yeGFP, AAA071; pTEF_luxO*-yeGFP + VP48-NLS-luxR, AAA070; pTEF_luxO*-yeGFP + luxR). In this setup, the VP48-fused construct again acted as an activator, whereas the construct without VP48 functioned as a repressor (Fig. S4). These results demonstrate that transcriptional output direction can be switched by inclusion or omission of the activation domain, but do not support a conditional dual-function

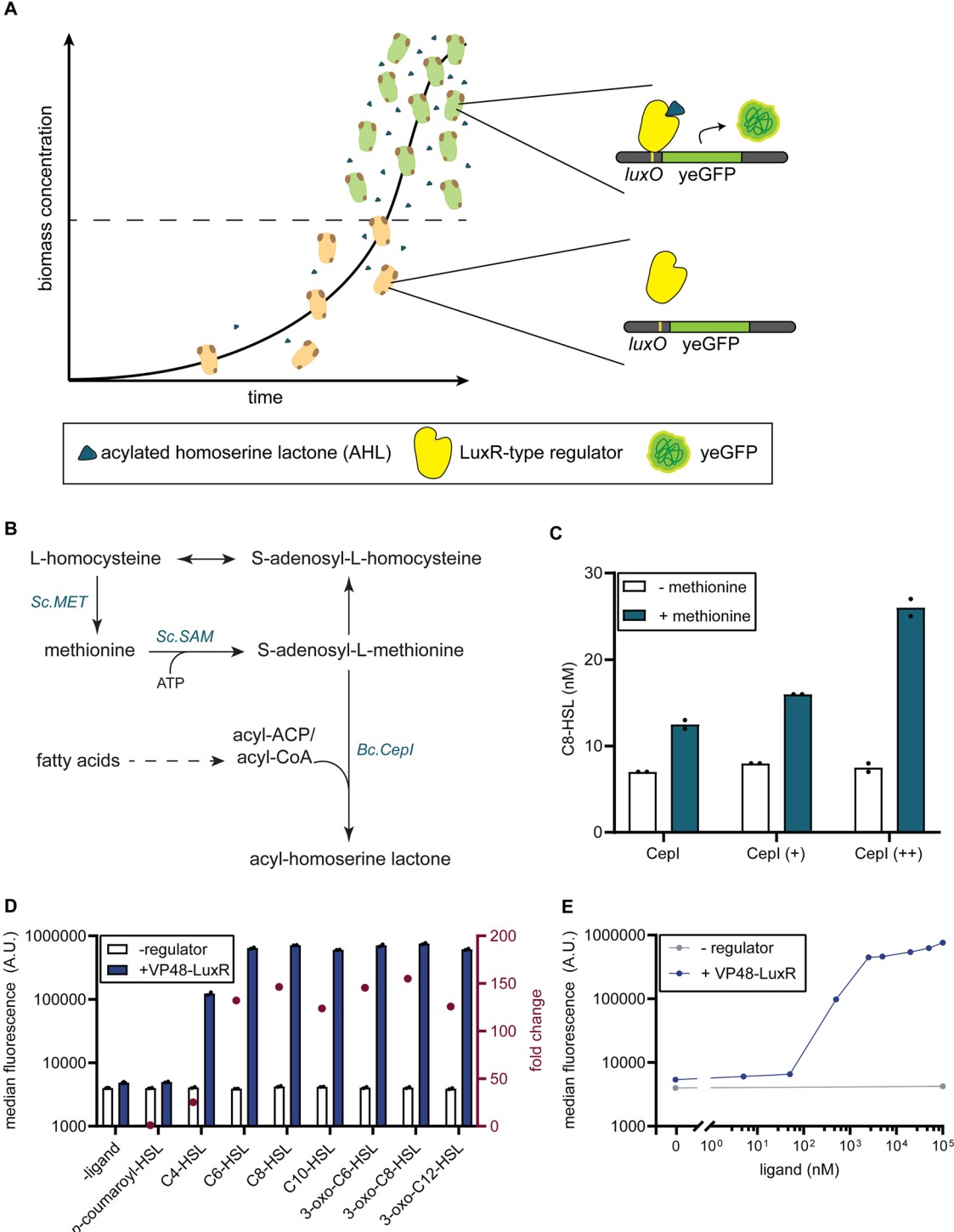

**Fig. 1 | Development of acylated homoserine lactone (AHL)-induced expression system in yeast. A** Schematic overview of AHL-dependent – QS-controlled regulation. The concentration of AHLs increases with an increase in biomass. After a certain threshold biomass concentration (ie. AHL concentration) is reached (indicated by the arbitrary dashed line), it modulates gene expression via interaction with a LuxR-type regulator and operator sequence (*luxO*). In this example, the expression of a fluorescent protein (yeGFP) is controlled. **B** Overview of the metabolic pathway involved in the formation of AHLs. **C** Production of C8-HSL established in yeast strains (AAA063; CepI, AAA113; CepI (+), AAA111; CepI (++)) expressing a *Bc.cepI* expression cassette alone (indicated by CepI) or with an overexpression cassette for *Sc.SAM2* and *Sc.MET6*. For the upregulation of *SAM2* and *MET6*, either

an expression cassette was introduced driven by p*ACT1* and p*PGI1* (indicated by (+)) or an expression cassette driven by p*TEF1* and p*PGK1* (indicated by (++)). Yeast strains were grown for 48 h in duplicate aerobic shakeflasks on fed-batch EnPump medium. **D** yeGFP fluorescence levels 6 h following supplementation of 0 or 100 μM ligand tested for strains AAA019 (-regulator) and AAA036 (+regulator) in duplicate cultures. Red dots indicate fold change in yeGFP fluorescence between 0 μM and 100 μM treatment. **E** Dose-response curve of yeGFP fluorescence levels 6 h following supplementation of 0–100,000 nM C8-HSL, tested for strains AAA019 (-regulator) and AAA036 (+regulator) in duplicate cultures. A.U.: arbitrary units. *Bc: Burkholderia cenocepacia.* Individual data points are shown.

**Fig. 2 | Development of growth-based screening using AmdS as (counter-) selectable marker.** Growth curves of batch cultures of *S. cerevisiae* strains expressing *Nd.amdS* from a minimal *GAL*-core promoter equipped with 5x*luxO* (red) (ACA004), from the strong p*TEF1* (green) (ACA002) and from a minimal *GAL*-core promoter equipped with 5x*luxO* controlled by VP48-LuxR (yellow) (ACA019) (**A**) on counter-selection medium (SMD supplemented with fluoro-acetamide (F-Ac), (**B**) and selection medium (SMD without nitrogen source supplemented with acetamide) supplemented with 5 µM 3-oxo-C6-HSL were obtained by a plate reader. Representative data of duplicate cultures are shown. **C** The pipeline workflow for enrichment of functional biosensors from a mixture of genetically different yeast strains consists of growing consecutively on counter- followed by (at least) two transfers on selection medium. **D** This workflow was tested in duplicate using the same three strains and using 5 µM 3-oxo-C6-HSL as ligand. The abundance of each strain was tracked using colony PCR of 24-32 single colonies after each transfer. *Nd.*: *Aspergillus nidulans*.

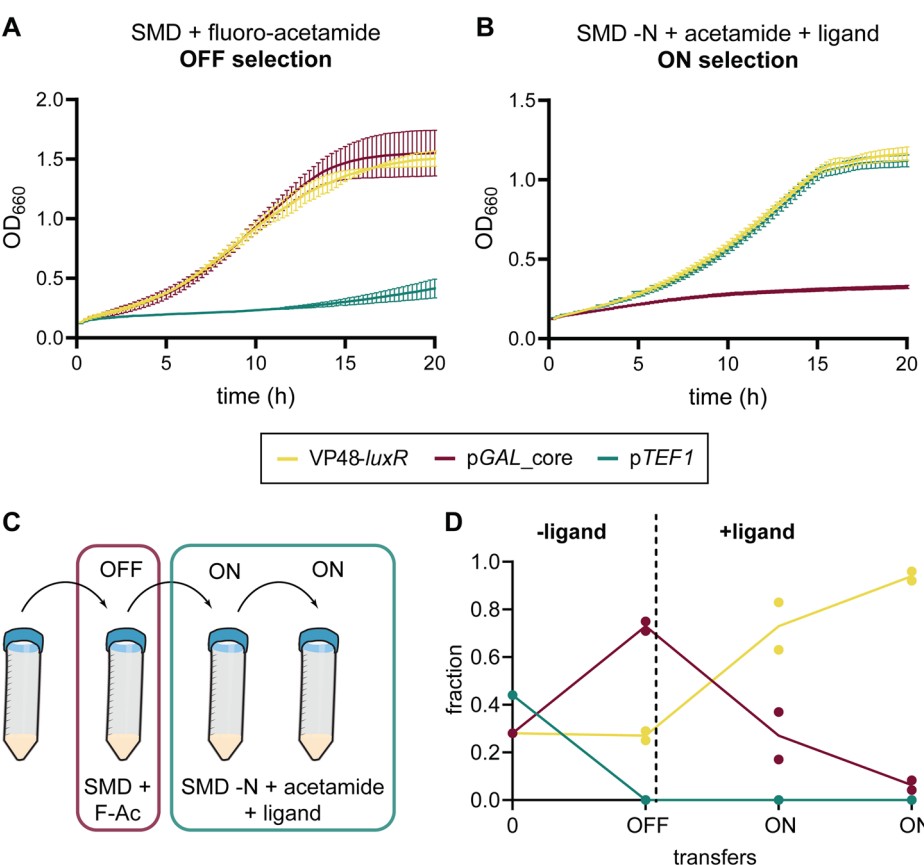

system. Overall, LuxR with an activation domain in combination with p*GAL*_core-5x*luxO* performed best for activation. This design responded to different chain lengths 3-oxo acyl-HSLs as well as to straight-chain acyl-HSLs of various lengths, including C8-HSL, with biosensor-dependent fold-changes in yeGFP expression reaching up to 155-fold (Fig. 1D). However, the operational range of the biosensor (≥ 500 nM) (Fig. 1E) did not correspond with our established C8-HSL production (5–30 nM) (Fig. 1C) and would, by design, therefore not be useful to complete the quorum-sensing circuit.

## Acetamidase can be used for growth-based (counter-) selection of biosensors

To make a QS system based on prokaryotic LuxR in yeast, the sensitivity of the transcriptional activator would need to be improved. In allosterically regulated proteins, directed evolution is often used for developing transcriptional regulation based on prokaryotic regulators, including optimization of sensitivity[5,32,36]. While several studies have used fluorescence-assisted cell sorting (FACS) to select specific phenotypes out of a pool of mutants[5,36], we decided to test whether the *Aspergillus nidulans amdS* gene (encoding an acetamidase (AmdS)) could be employed as a (counter-) selectable marker for growth-based screening. Briefly, AmdS converts acetamide to acetate and ammonia, thereby allowing the host to grow on acetamide as the sole nitrogen (or carbon) source[37]. In contrast to auxotrophic markers, *amdS* is a dominant gain-of-function marker requiring a higher abundance of the enzyme for a fast growth rate, therefore allowing for a greater dynamic range to be covered by the marker. Additionally, the acetamide homologue fluoroacetamide can be used for counter-selection since it is converted by AmdS to the toxic product fluoroacetate. If functional, such a system allows for ease of use and cost reduction in terms of instrumentation. As the counter-selection and the selection can be performed using the same marker, enrichment for loss-of-function mutations (which can potentially occur during the counter-selection) would be limited since these mutants would not be enriched during the selection round.

While AmdS is already routinely applied to facilitate screening of genetic modifications[38–41], its application for high-throughput growth-based directed evolution has, to the best of our knowledge, not been demonstrated before.

At first, *amdS* was fused to yeGFP with a $(G_4S)_3$-linker, to allow for growth-based selection and simultaneous phenotype analysis. The *TEF1*-promoter (p*TEF1*) was used to simulate the maximum ON-state of a biosensor and the minimal *GAL* core-promoter equipped with 5 *luxO* operator sites (p*GAL*_core) as the OFF-state. SMD medium supplemented with fluoro-acetamide (F-Ac) was used as counter selection, while SMD without nitrogen source (SMD-N) supplemented with acetamide, was used as selection medium (which will be supplemented with the ligand). Importantly, the medium cannot be supplemented with amino acids to complement auxotrophies of your yeast, since these can be used as an alternative nitrogen source. We therefore switched to CEN.PK110-10C-derived yeast strains (containing Cas9-cassettes on a plasmid carrying a *HIS3*-marker) when using *amdS*.

To be able to enrich for functional biosensors, there needed to be a substantial difference in growth rate between the ON and OFF states on both the counter-selection medium as well as the selection medium. We therefore started by testing the strains (ACA007; p*GAL*_core-*amdS*-yeGFP, ACA013; p*TEF1*-*amdS*-yeGFP) in a plate reader, to have an estimation of the relative growth rates. Using 5 g L⁻¹ F-Ac, an approximate 33-50% reduction of the growth rate was observed for the always ON-state compared to the always OFF-state (Fig. S5). Increasing the concentration to 10 and 20 g L⁻¹ F- Ac did not further reduce the growth rate. In parallel, *amdS* without the yeGFP fusion was also tested (strains ACA002; p*TEF1*-*amdS*, ACA004; p*GAL*_core-*amdS*), and displayed a 78%, 80% and 82% reduction in growth rate at 5, 10 and 20 g L⁻¹ F-Ac, respectively (Figs. 2A, S5). The fusion of *amdS* with yeGFP appears to interfere with the activity of AmdS. This is corroborated by the growth curves obtained by growing the strains on SMD -N + acetamide, which displayed an approximately 20% reduction in growth rate for the AmdS-yeGFP fusion, compared to AmdS (Fig. S6A).

Importantly, for both AmdS-yeGFP fusion as well as AmdS, on the selection medium we observed an approximate 55% and 62% reduction in growth rate between the ON-state and the OFF-state, and the OFF-state strains are virtually non-growing (Figs. 2B, S6A). To get a better understanding about cross-feeding[42,43] using *amdS* as selection marker, we cultured the ON and OFF-state together and tracked the abundance of each strain (ACA013; pTEF1-*amdS*-yeGFP and ACA008; pGAL_core-*amdS*-mKate2) during two consecutive transfers (Fig. S6B). Based on this data, we concluded that *amdS* can be used as a selection marker with minimal cross-feeding. Since the counter-selection is slightly more stringent when *amdS* is expressed as an individual ORF compared to the fusion version (82% reduction compared to 50% reduction in growth rate), we decided to move forward by expressing *amdS* independently instead of as a fusion.

We started with testing the complete workflow, by sequentially growing a mixture of strains on the counter-selection medium, followed by the selection medium (Fig. 2C). The mixture consisted of the yeast strain mimicking the always OFF-state (ACA004: pGAL_core), the always ON-state (ACA002: pTEF1) and a yeast with a functional biosensor which can toggle between the ON and OFF-state (ACA019: VP48-LuxR). No ligand was added to the counter-selection medium, while 5 µM 3-oxo-C6-HSL was added to the selection medium. The abundance of each strain was tracked by plating the culture before each transfer and quantifying genotypically. Indeed, we were able to enrich for a functional variant out of a pool that mimics non-functional biosensors (Fig. 2D), while there was no enrichment when growing for 3 consecutive transfers on non-selective medium (Fig. S7). For application in library screening, we decided to integrate yeGFP driven by the hybrid promoter in a separate locus, to be able to directly verify biosensor behaviour in the population during the different transfers (Fig. 3B).

## Increasing sensitivity of LuxR by directed evolution strategies

After establishing a workflow for screening mutant libraries, we tested different activation domains in combination with LuxR and found that, consistent with previous studies[44], we found Med2 to be among the most potent activators and observed that VP48 also drove strong activation (Fig. S8). By contrast, VP16 did not induce any detectable expression, while Gal4 activation domain (AD) resulted in the mildest activation (Fig. S8). Given that overly strong activation could lead to unintended effects, such as system saturation, we chose to proceed with Gal4_AD to allow for more controlled and tunable gene expression in our system. To allow for any synergistic effects between variant regulator and activation domains, the whole open reading frame of *luxR*, as well as the *GAL4*_AD was subjected to error-prone PCR (epPCR) (Fig. 3A). After optimizing the transformation protocol (see Methods), a mutant library of approximately 2.7 million mutants was obtained.

The library was grown on counter-selective medium (SMD 20 F-Ac) followed by two transfers on selective medium (SMD-N + acetamide) supplemented with 5 nM or 50 nM C8-HSL. On the second transfer on selective medium, no growth was observed after 3 days on the selective medium supplemented with 5 nM C8-HSL, but mutants were able to grow on the selective medium supplemented with 50 nM C8-HSL. Analysing the population (T1) on the flow cytometer indicated a subpopulation that was more fluorescent (Fig. 3C, S9A), corresponding to 32% of the total culture. We decided to perform three more transfers on the selective medium, obtaining populations T2, T3 and T4. For each population T1-4, we analysed the population as a whole as well as 48 single colonies, using flow cytometer. Using consecutive transfers on the selection medium, the abundance of the more fluorescent subpopulation could be increased to 88%, with 94% of the randomly picked colonies displaying an improved phenotype (Figs. 3C, T4) and no false-positives. Alternatively, we found that this 32% subpopulation in T1 could also be easily enriched using FACS-selection (Fig. S9B) as well as by manually picking the most fluorescent colonies when plating on YPD supplemented with ligand (Fig. S9C), resulting in 98% and 65% of the picked colonies displaying an improved phenotype, respectively. The best performing mutants (fold change >10) were genotyped by sequencing.

## Identification of Gal4-LuxR-variants with increased sensitivity for acyl-HSLs

Analysing the genotypes of the best performing mutants, we identified 3 unique genotypes (Table 1). Regardless of the method for further selection, all 3 genotypes were found in the selected colonies. Notably, all selected mutants are mutated in residue N86 (i.e., N86I, N86K and N86Y). The N86K mutation has also been reported to increase sensitivity in *E. coli*, where it was suggested to play a role in changing conformation upon ligand binding[45].

We integrated the mutant regulators into strain AAA019 (pGAL_core-5x*luxO*-yeGFP) for phenotypic analysis. Since N86K mutation specifically has been reported previously in *E. coli*[45], additionally a biosensor strain carrying LuxR_N86K was constructed to assess the effect of this specific mutation. To evaluate the impact of these mutations, we performed dose-response assays using C8-HSL as well as C4-HSL, C6-HSL, C10-HSL, 3-oxo-C6-HSL, 3-oxo-C8-HSL and 3-oxo-C12-HSL (Figs. 4, S10).

Analysis of the dose-response curves showed that the N86K mutation led to a notable increase in sensitivity for C8-HSL compared to the control regulator (Fig. 4B). However, the three mutants identified from the mutant library displayed even greater sensitivity, with genotypes gen1 and gen2 displaying the highest sensitivity. These findings suggest that while the N86K mutation improves sensitivity, additional mutations present in the other 3 mutants further amplify this effect. Moreover, this effect is not specific for C8-HSL, since all mutants displayed increased sensitivity towards other acyl-HSLs as well (Fig. S10). Lastly, genotypes gen1 and gen2 displayed a 2 and 3.5-fold increase in fluorescence in the presence of 10 nM C8-HSL, respectively, indicating induction in the range of C8-HSL production previously established in yeast in this study (Fig. 1C).

## Implementing orthogonal quorum-sensing circuits in *Saccharomyces cerevisiae*

QS-circuits typically consist of a sender module, responsible for acyl-HSL production, and a sensor module, responsible for acyl-HSL detection. With C8-HSL production established and the identification of sensitive mutants completed, we finalized the construction of QS circuits. To achieve this, we integrated the *luxR*-variants together with the hybrid promoter (QS-sensor module) into strain AAA111 (CepI (++)), which already contained expression cassettes for *Bc.CepI*, *Sc.SAM2* and *Sc.MET6* (QS-sender module) (Fig. 4C). Biomass and fluorescence were measured over time in a plate reader for quadruplicate cultures of the resulting strains grown in fed-batch EnPump medium supplemented with methionine. Our results confirmed that combining both modules resulted in the first successful implementation of an AHL-based QS-controlled yeGFP expression system in yeast, characterized by delayed self-induced yeGFP expression (Fig. 4D). For comparisons between the different circuits, induction was defined as a 1.8-fold increase in fluorescence normalized to OD and activity was defined as the time required to reach this induction level, further normalized to the exact induction level at that timepoint (see Methods). Consistent with the dose-response curves (Fig. 4B), strains carrying gen1 and gen2 genotypes (Table 1) induced yeGFP expression earlier in the cultivation (at 14.5 and 13.6 h, respectively) than genotype gen3 and N86K (at 17.8 and 20.4 h, respectively) (Fig. 4D). Circuits based on gen1 and gen2 thereby showed $50 \pm 2\%$ and $57 \pm 1\%$ greater activity, and reached 3.6- and 2.2-fold higher maximum induction levels (induction levels of $6.7 \pm 0.2$ and $4.1 \pm 0.1$, for gen1 and gen2 respectively) than N86K, while gen3 exhibited $19 \pm 2\%$ increase in activity relative to N86K and reached a 1.5-fold higher level of induction (induction level of $2.7 \pm 0.1$) (Fig. 4D). To further investigate the impact of C8-HSL production levels, we integrated *luxR*_gen1 into strains AAA063 (CepI) and AAA113 (CepI (+)), which were previously shown to produce 1.9-fold and 1.5-fold less C8-HSL than AAA111 (CepI (++)), respectively (Fig. 1C). Strains AAA151 (CepI (++) + Gal4-LuxR_gen1), AAA169 (CepI (+) + Gal4-LuxR_gen1) and AAA168 (CepI + Gal4-LuxR_gen1) were grown in medium supplemented with and without methionine. As expected, activity was lower for the QS-circuits in which less C8-HSL was produced, with a marginal 30% higher level of induction

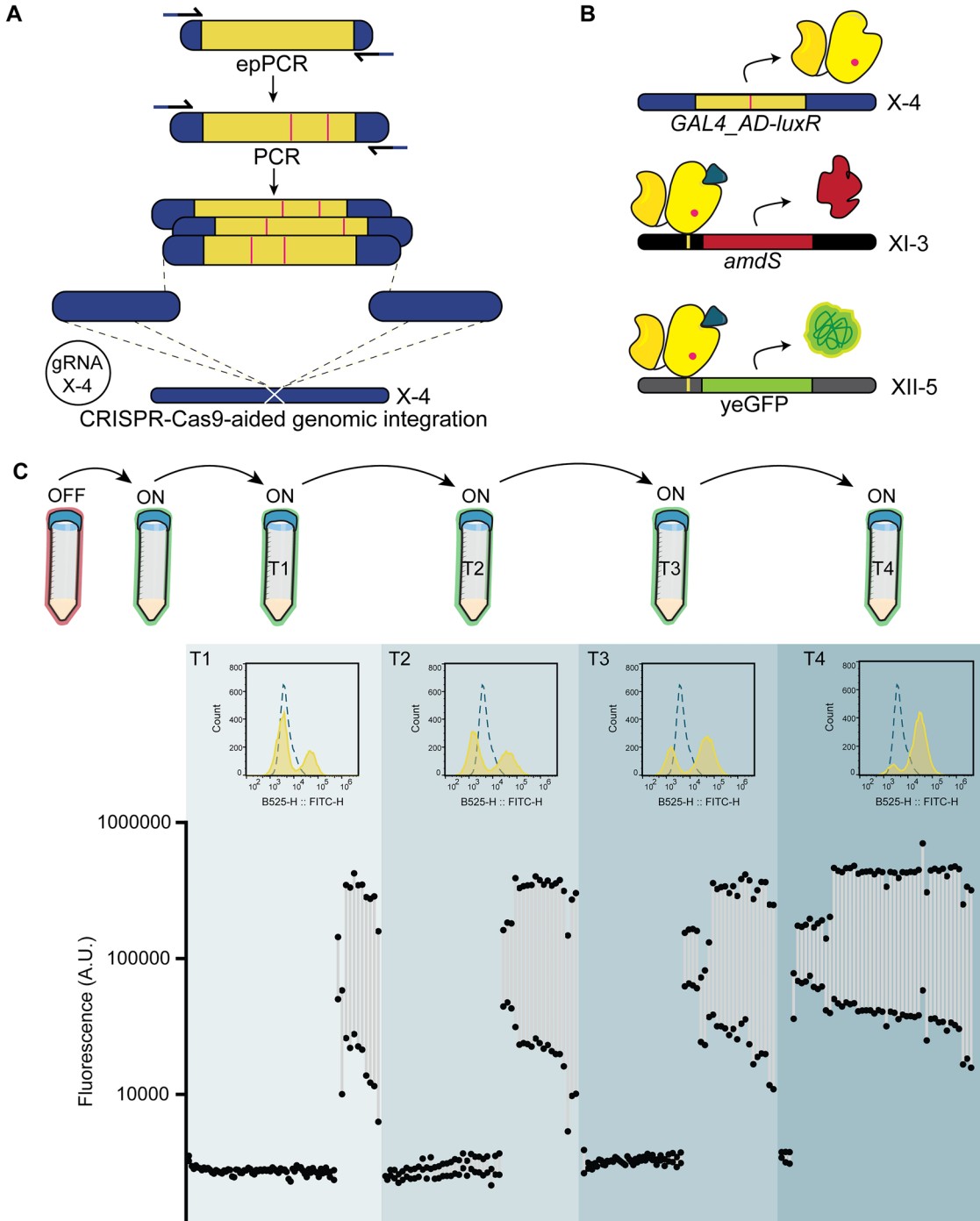

**Fig. 3 | Construction and screening of a mutant library of *GAL4*_AD-*luxR*.**
**A** Error-prone PCR (epPCR) was performed on the *GAL4*_AD-*luxR* sequence using primers containing 60 bp overlaps with the promoter and terminator regions. A second round of regular high-fidelity PCR was used to amplify the mutants and extend the overlaps by an additional 60 bp (for a total of 120 bp of homology). The promoter and terminator sequence, each containing 400 bp flanks homologous to the integration site, were amplified using regular high-fidelity PCR. The three fragments (promoter, mutant *GAL4*_AD-*luxR* and terminator) were assembled into the X-4 integration site by CRISPR-Cas9-mediated homologous recombination.

**B** Schematic of the resulting yeast strain(s), which contained *amdS* and yeGFP controlled by p*GAL*_core_5x*luxO*. **C** The mutant library was grown on transformation selection media (SC -HIS + NAT) for 3 days, after which cells were transferred to counter-selection media (OFF; SMD + F-Ac) for 1 day. Subsequently, cells were transferred up to 5 times on selection media (ON; SMD -N + acetamide), containing 50 nM C8-HSL. Cultures T1-T4 were analysed on the flow cytometer and plated for single colonies after each transfer. 48 single colonies from each culture were analysed by flow cytometry after 6 h growth with or without ligand, to determine the dynamic range of GFP expression. A.U.: arbitrary units.

reached for strain AAA168 compared to AAA169 ($2.8 \pm 0.1$ and $2.2 \pm 0.1$, respectively) (Fig. S11B). Moreover, the addition of methionine did not significantly influence the timing of yeGFP induction and or activity of the QS-circuit (Fig. S11C), and it was therefore omitted in subsequent experiments.

## Removing activation domain from LuxR inverses mode-of-action of regulator

To expand the versatility of QS-controlled regulatory systems in yeast and enable both QS-controlled activation and repression, we decided to build on our initial observation that LuxR lacking VP48 acted as a repressor when a

*luxO*-type operator was positioned downstream of the TATA box in the *TEF1* promoter. To test whether repression could be strengthened, we replaced this operator sequence with the exact *luxO* sequence that was used in the p*GAL*_core-5x*luxO* design, resulting in strain AAA096. Additionally, we integrated *luxR* (AAA170), *luxR*_gen1 (AAA203) and *luxR*_N86K (AAA171). In the presence of the ligand, indeed, a 2 to 3-fold reduction of yeGFP expression was observed (Fig. 5B). Moreover, both the gen1 mutation set and the N86K substitution alone increased sensitivity towards C8-HSL in this new context. We tried to further improve the dynamic range by increasing the operator binding sites from 1 to 2, which enhanced LuxR-dependent repression in studies performed in *E. coli*[34]. This lowered the basal activity of the system but did not further increase the dynamic range in yeast (Fig. S12).

## QS-controlled repression of *FAS1* improves aloesone production

QS-controlled regulation can be very powerful to autonomously divert metabolic fluxes towards a specific pathway, later in the production process. This can be beneficial in systems where biosynthesis and production compete for the same precursors. To illustrate this, we decided to regulate an anabolic production pathway using our established QS-tools. As a testbed, we decided on the production of aloesone, a bioactive compound known for its pharmaceutical properties[46]. This compound is produced from malonyl-CoA and acetyl-CoA by a polyketide synthase[47]. Malonyl-CoA and acetyl-CoA are also precursors required in the fatty acid synthesis (FAS), as well, which is therefore in direct competition with aloesone production (Fig. 5C). Previous studies have established that regulation of *FAS1*[48] can be targeted to improve malonyl-coa-derived products[49–52]. Building on these strategies, we replaced p*FAS1* with p*TEF1*_*luxO*, to allow for QS-controlled *FAS1* gene

expression. We further introduced *Aa.PKS3* from *Aloe vera* (strain AAA135; PKS3), QS-sender module (strain AAA206; PKS3, CepI (++)), QS-sensor module (strain AAA200; PKS3, LuxR-gen1) and finally the complete QS-circuit (strain AAA202; PKS3, LuxR-gen1, CepI (++)). Strains were tested in Biolector plates in glucose fedbatch mode, with exponential feeding and pH controlled to stay above 5.6. All strains exhibited similar growth profiles under these conditions (Fig. S13). End-point measurements were performed to obtain the relative estimated abundance of aloesone (Fig. S14) and C8-HSL in each of the strains. Production of signalling molecule C8-HSL did not change the production of aloesone, indicating that the metabolic burden from production of the autoinducer is neglectable (Fig. 5D; PKS3+ CepI (++)). Introduction of only *luxR*-gen1 marginally increased production in strain (Fig. 5D; PKS3 + LuxR-gen1), likely by lowering the basal expression of *FAS1*. Introduction of the full QS-circuit significantly increased production of aloesone by 51% reaching 24.4 ± 1.7 nM, likely by enhancing the supply of malonyl-CoA for aloesone production (Fig. 5D; PKS3 + LuxR-gen1 + CepI (++)).

## Discussion

The AHL-based QS-system is a widely studied mechanism of cell-cell communication in Gram-negative bacteria[24] and this study represents the first successful engineering of an AHL-based QS-system in yeast. A key limitation of our current system is that the tested synthases exhibited limited activity in yeast, necessitating overexpression of *SAM2* and *MET6* to boost C8-HSL production. This overexpression imposes a measurable growth defect on the host and may compromise circuit stability. Selecting or engineering higher-activity synthases could improve QS circuit performance while minimizing additional metabolic stress on the host. Functional expression of an AHL synthase therefore, remains a major challenge. Most LuxI-family AHL synthases utilize acyl-ACPs as acyl donors, and acyl-CoA to a much lesser extent[16,18,53]. However, unlike many prokaryotes, yeast does not have freely available acyl-ACP, as it exists only as part of the multi-functional fatty acid synthase type 1 (FAS I) complex[54–56]. We hypothesize that this difference prevents direct donation of an acyl-group to LuxI-type enzymes, potentially limiting their activity in yeast. One possible strategy to overcome this limitation would be the introduction of bacterial FAS to provide free acyl-ACP, although this is expected to impose a significant metabolic burden and growth defects. Alternatively, exceptions have been reported of synthases that are more efficient in utilizing an acyl-CoA as the

### Table 1 | Overview of mutations found in Gal4-NLS-luxR

| Location | gen1 | gen2 | gen3 |
|---|---|---|---|
| Gal4_AD | N24K, P41 (CCA → CCG), N46D, T92S, V98 (GTA → GTT), K114E | G8V | F51 (TTC → TTT), G88E |
| NLS | | K120N | |
| LuxR | N5D, V36 (GTT → GTC), N86I, S164Y, D182 (GAT → GAC) | N86K, K104E, M135V | K53E, N86Y, I119V |

**Fig. 4 | Characterization of Gal4-LuxR-mutants and implementation into QS-circuits. A** Structural mapping of mutations found in gen1 (green), gen2 (pink) and gen 3 (blue) in LuxR. The structure of LuxR was obtained via Alphafold3[62] prediction. **B** Dose-response curves of yeGFP fluorescence levels 6 h following supplementation of 0–10,000 nM C8-HSL. Tested for *S. cerevisiae* strains AAA036 (- regulator), AAA101 (+regulator), AAA156 (+ gen1), AAA157 (+ gen2), AAA158 (+gen3), AAA107 (+N86K). **C** Improved biosensors were implemented into C8-HSL-producing strains consisting of an overexpression module of *Sc.SAM2* and *Sc.MET6* and *cepI* (CepI (++)). **D** Fluorescence and $OD_{660}$ were monitored in fluorescence plate reader of *S. cerevisiae* strains AAA148 (CepI (++) + N86K), AAA149 (CepI (++) + gen3), AAA150 (CepI (++) + gen2), AAA151 (CepI (++) + gen1) and control strain AAA189 (CepI (++) – regulator). Cultures were grown on fed-batch medium to analyse the autonomous delayed induction of GFP. A.U.: arbitrary units. *Bc: Burkholderia cenocepacia*.

**Fig. 5 | Establishing of repressive behaviour of LuxR-based biosensor and implementation into a QS-controlled circuit to increase production of aloesone. A** Removal of the activation domain and insertion of *luxO* sequence downstream of the TATA-box in p*TEF1* (p*TEF1*_luxO) to convert LuxR into a transcriptional repressor in the presence of ligand. **B** Dose-response curves of yeGFP fluorescence levels following a 24 h pre-incubation with 5–10,000 nM C8-HSL prior to transfer to fresh medium with supplementation for 6 h with 5–10,000 nM C8-HSL with *S. cerevisiae* strains AAA097 (-LuxR), AAA170 (+LuxR), AAA171 (+LuxR-N86K) and AAA203 (+LuxR-gen1). **C** Schematic overview of the production of aloesone by *Aa.PKS3* from acetyl-CoA and malonyl-CoA, emphasizing the resource competition with fatty acid synthesis. pFAS1 is replaced with p*TEF1*_luxO. **D** End-point determination of C8-HSL production and aloesone production from aerobic fed-batch cultivation performed in flowerplates analysed by biolector (n = 4–5) with strains AAA135 (PKS3), AAA206 (PKS3, CepI (++)), AAA200 (PKS3, LuxR-gen1) and AAA202 (PKS3, LuxR-gen1, CepI (++)). A.U.: arbitrary units. *Aa.*: *Aloe vera.* *Bc.*: *Burkholderia cenocepacia.* *** indicates *p* < 0.001.

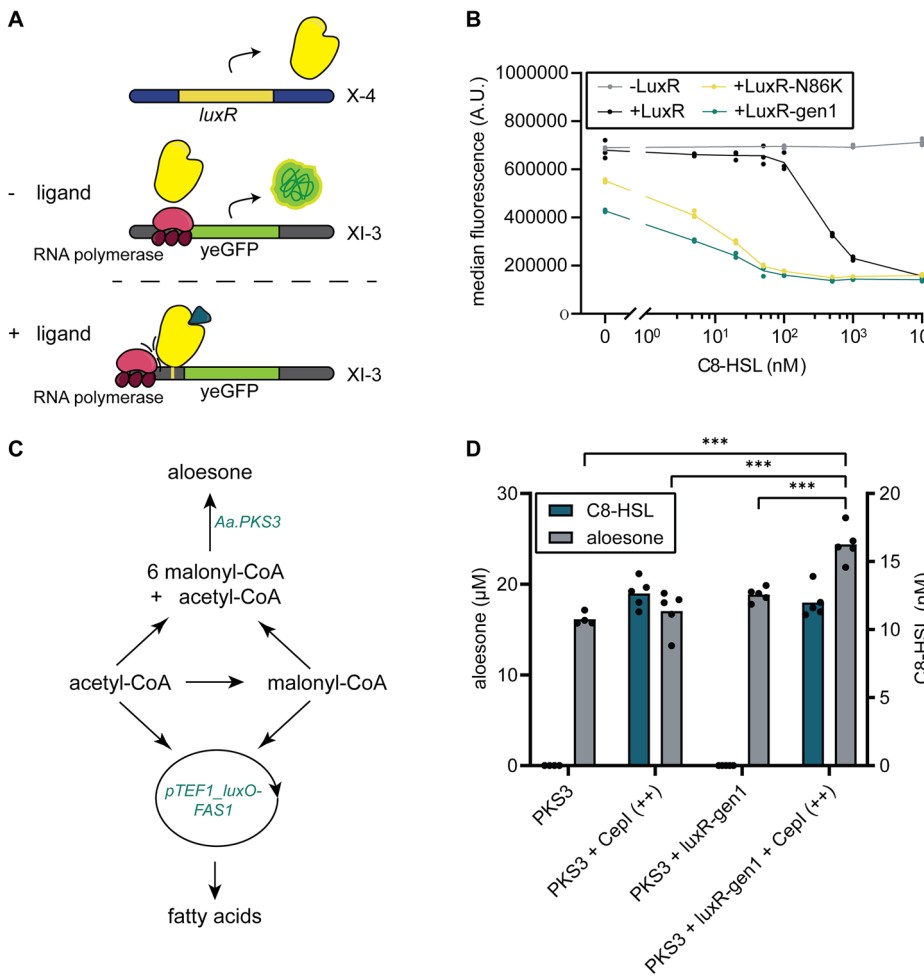

acyl-group donor[57–61]. Selecting acyl-CoA–dependent synthases could potentially improve QS circuits without requiring precursor-boosting cassettes, minimizing metabolic stress. Future work should also assess the long-term stability of the system during extended fermentations to ensure consistent performance.

A second challenge was posed by the low concentration of C8-HSL, requiring a highly sensitive detection system. From the Alphafold3[62] structures of all 3 mutants compared to wildtype (WT) no change was seen in the ligand binding mode of C8-HSL after docking (Fig. S15). However, for gen 2 (K120N, N86K, K104E, M135V), we found that M135 and K104 are in the loop in front of the entrance of the tunnel (Fig. S16). Possibly, these mutations could have an effect on ligand binding, especially for 3-oxo-AHLs. In line with this observation, while gen2 displays the highest sensitivity for all tested straight-chain AHLs, gen1 and 3 are more sensitive for the tested 3-oxo-AHLs (Fig. S10). Additionally, the mutant versions also seem to have an increased baseline expression of the target gene. The observed trend may reflect an inherent trade-off between increasing biosensor sensitivity (lowering activation threshold) and maintaining tight OFF-state, as these properties are often inversely related[63]. Further experimental work and molecular dynamics simulations would be needed in the future to fully deconvolute the individual role of each mutation identified, and any mechanistic effects associated.

In our hands, fusion of LuxR with the strong VP48 activation domain abolished repression and converted the construct into an activator (Fig. S3). In this study, we therefore employed a separate design for repression, using LuxR without an activation domain to ensure robust and consistent repression. However, alternative activation domains with lower activity may strike a better balance, potentially enabling LuxR to function as both activator and

repressor in the same strain. Such a system could, for example, delay activation of toxic gene products and delay repression of biosynthetic processes. Developing dual-purpose regulators remains a promising strategy for coordinated control of multiple genes and pathways within a single chassis.

Still, despite the current limitations, the implementation of an AHL-based quorum-sensing system establishes a truly orthogonal QS system in yeast. Such a system enables cell density-dependent regulatory control, as demonstrated by its application in aloesone production. Additionally, it can facilitate synthetic microbial consortia by coordinating interactions between different yeast strains, allowing for division of labour in metabolic processes. This key application could be extended to cross-species communication as well, enabling engineered yeast to interact with bacteria in co-cultures for improved bioproduction or cooperative behaviours. The application of AHL-based QS-circuits could be envisioned in interspecies co-cultures to, for example, help optimize microbial production of plant secondary metabolites[64]. Precursors derived from aromatic amino acids, phenylalanine and tyrosine, are efficiently synthesized in prokaryotes[65–67]. On the other hand, P450 enzymes, catalyzing the crucial steps in the synthesis of bioactive compounds, are poorly expressed in hosts like *E. coli* but highly functional in yeast[66,68]. With such cross-species cooperation, challenges related to differences in growth rate, as well as required precursors that compete with biosynthesis, could be addressed by applying AHL-based QS regulatory circuits. This could prove instrumental in developing bioprocesses.

## Methods
### Strains and growth media
The *Saccharomyces cerevisiae* strains used in this study are listed in Table S1 and are derived from the CEN.PK-lineage[69]. CEN.PK110-10C (MAT-a

URA3 LEU2 TRP1 his3) was used to construct all strains for the screening work (using *amdS*), while all other strains were derived from CEN.PK2-1C (MAT-a ura3 his3 leu2 trp1). The yeast strains were routinely cultivated at 30 °C in synthetic complete medium (SC) (6.7 g L$^{-1}$ yeast nitrogen base without amino acids, 1.62 g L$^{-1}$ yeast synthetic drop-out medium supplement without leucine, 0.2 g L$^{-1}$ leucine, 20 g L$^{-1}$ glucose, pH set to 5.6 with 2 M KOH). Selection was performed in synthetic complete medium lacking histidine (SC -HIS) or lacking both histidine and uracil (SC -HIS -URA) (6.7 g L$^{-1}$ yeast nitrogen base without amino acids, 1.92 g L$^{-1}$ yeast synthetic drop-out medium supplement without histidine or 1.39 g L$^{-1}$ yeast synthetic drop-out medium supplement without histidine, leucine, tryptophan and uracil supplemented with 0.2 g L$^{-1}$ leucine and 0.07 g L$^{-1}$ tryptophan, 20 g L$^{-1}$ glucose, pH set to 5.6 with 2 M KOH, 2% (w/v) agar in case of plates). Selective cultivation using 100 mg L$^{-1}$ nourseothricin (clonNAT, Werner BioAgents) was done using synthetic media containing monosodium glutamate (SMG) (1.7 g L$^{-1}$ yeast nitrogen base without amino acids and ammonium sulfate, 1 g L$^{-1}$ monosodium glutamate, 1.92 g L$^{-1}$ yeast synthetic drop-out medium supplement without histidine or 1.39 g L$^{-1}$ yeast synthetic drop-out medium supplement without histidine, leucine, tryptophan and uracil supplemented with 0.2 g L$^{-1}$ leucine and 0.07 g L$^{-1}$ tryptophan, 20 g L$^{-1}$ glucose, pH set to 5.6 with 2 M KOH, 2% (w/v) agar in case of plates). Synthetic minimal medium (3.0 g L$^{-1}$ KH$_2$PO$_4$, 0.5 g L$^{-1}$ MgSO$_4$·7H$_2$O, 5.0 g L$^{-1}$ (NH$_4$)$_2$SO$_4$, 1 ml L$^{-1}$ trace elements and vitamins[70]) supplemented with 20 g L$^{-1}$ glucose (SMD) was used for screening the libraries. Trace elements contained 15 mg L$^{-1}$ EDTA, 4.5 mg L$^{-1}$ ZnSO$_4$·7H$_2$O, 0.3 mg L$^{-1}$ CoCl$_2$·6H$_2$O, 1 mg L$^{-1}$ MnCl$_2$·4H$_2$O, 0.3 mg L$^{-1}$ CuSO$_4$·5H$_2$O, 4.5 mg L$^{-1}$ CaCl$_2$·2H$_2$O, 3 mg L$^{-1}$ FeSO$_4$·7H$_2$O, 0.4 mg L$^{-1}$ NaMoO$_4$·2H$_2$O, 1 mg L$^{-1}$ H$_3$BO$_3$ and 0.1 mg L$^{-1}$ KI. Vitamin solution contained 0.05 mg L$^{-1}$ biotin, 1 mg L$^{-1}$ calcium pantothenate, 1 mg L$^{-1}$ nicotinic acid, 25 mg L$^{-1}$ inositol, 1 mg L$^{-1}$ thiamine HCl, 1 mg L$^{-1}$ pyridoxine HCl and 0.2 mg L$^{-1}$ para-aminobenzoic acid. Fluoroacetamide (final concentration of 20 g L$^{-1}$) was added to SMD for the OFF-selection, while (NH$_4$)$_2$SO$_4$ was replaced by 6.6 g L$^{-1}$ K$_2$SO$_4$ and 0.6 g L$^{-1}$ filter-sterilized acetamide for the ON-selection. Extra-buffered fed-batch medium contained 5.0 g L$^{-1}$ KH$_2$PO$_4$, 0.5 g L$^{-1}$ MgSO$_4$·7H$_2$O, 14.4 g L$^{-1}$ (NH$_4$)$_2$SO$_4$, 1 ml L$^{-1}$ trace elements, 1 ml L$^{-1}$ vitamins, 0.125 g L$^{-1}$ histidine, 0.5 g L$^{-1}$ leucine, 0.075 g L$^{-1}$ tryptophan, 0.15 g L$^{-1}$ uracil, 20 g L$^{-1}$ glucose and 40 g L$^{-1}$ EnPump 200 substrate (Enpresso, Berlin, Germany). Where mentioned, methionine was added to a final concentration of 1 g L$^{-1}$. Fed-batch was started with 8 mL L$^{-1}$ of enzyme mix. Feed medium for Biolector fed-batch cultivations contained 18 g L$^{-1}$ KH$_2$PO$_4$, 3 g L$^{-1}$ MgSO$_4$·7H$_2$O, 45 g L$^{-1}$ (NH$_4$)$_2$SO$_4$, 12 ml L$^{-1}$ trace elements, 6 ml L$^{-1}$ vitamins, 1.0 g L$^{-1}$ histidine, 5.0 g L$^{-1}$ leucine, 0.8 g L$^{-1}$ tryptophan, 1.5 g L$^{-1}$ uracil, 160 g L$^{-1}$ glucose. The start medium was prepared by diluting the feed medium 4 times in sterile Milli-Q.

For cloning and plasmid propagation, *Escherichia coli* strain DH5 was used, in Luria-Bertani (LB) medium containing 100 μg mL$^{-1}$ ampicillin or 25 μg mL$^{-1}$ chloramphenicol.

## Chemicals
Fluoroacetamide (Sigma-Aldrich, 128341-5 G) was dissolved in dH$_2$O to a final concentration of 200 gL$^{-1}$ and stored at 4 °C. Acylated homoserine-lactones C4-HSL (SML3427-10MG), C6-HSL (56395-10MG), C8-HSL (44558-10MG), C10-HSL (07028-10MG), 3-oxo-C6-HSL (K3007-10MG and K3255-25MG), 3-oxo-C8-HSL (O1764-10MG) and 3-oxo-C12-HSL (O9139-10MG) were purchased from Sigma-Aldrich and dissolved in 100% DMSO to a final concentration of 10 mM, aliquoted and stored at −20 °C. Final working concentrations in cultivations were 100 μM and lower, resulting in DMSO concentrations of below 1%.

## Plasmid construction
The plasmids used in this study are listed in Table S3 and relevant primers for plasmid construction in Table S2. The coding sequences of luxR, esaI, luxI and lasI were obtained from Addgene (Plasmids #165971[32], #47660[29], #73445[24], #73444[24]) as well as the core promoter with 5 operator sequences

pGAL1_5xluxO (plasmid #165977[32]). Codon-optimized versions of cepI, esaI, luxI and lasI, as well as pTEF1_luxO_105 were synthesized by GeneArt Thermo Scientific (sequences are shown in Table S5). Plasmid construction for integration fragments was performed by USER-cloning, using fragment-specific primers listed (Table S2) and Phusion U High-Fidelity DNA Polymerase (New England Biolabs) according to the manufacturer's instructions, to obtain overhangs suitable for USER-cloning. To obtain the pTEF1_2xluxO fragment with USER-overhangs, pAvA098 was amplified with AA34/AA201 and AA202/AA35. For the construction of single ORF cassettes, the backbone was PCR-amplified with MAD3/MAD4[71]. For the construction of plasmids with two ORFs, the backbone was linearized using SfaAI FD and BsmI FD (Thermo Scientific)[72]. Linearized backbone, promoter(s), ORF(s), terminator(s) and, where indicated, activation domain, were added in equimolar amounts with cutsmart buffer and USER-enzyme and handled according to the supplier's protocol. 2 μL of this mixture was used in the subsequent transformation with *E. coli* and the whole mixture was plated on selective medium. Plasmid encoding pFAS1 gRNA was constructed by PCR amplification and ligation[73] using the manually identified GTTTTAGAGCTAGAAATAGCAAGTTAAAATAAGGC as gRNA sequence and primers AA180 and phosphorylated primer TJOS-20[73].

## Yeast strains construction
Yeast strains were constructed using CRISPR-Cas9-mediated genome editing. A plasmid carrying Cas9, using HIS3 as a selection marker, was introduced into CEN.PK110-10C and CEN.PK2-1C and were stocked as ACA001 and AAA001, respectively and were used for all subsequent yeast transformations. In general, for each transformation one or two gRNA-plasmids were introduced alongside NotI-digested plasmids containing the integration cassette according to EasyClone protolcol[72] and as described in table S4. Exceptions include the introduction of pPGK1-GAL4_AD-luxR_N86K-tADH1 and pPGK1-luxR_N86K-tADH1, which were introduced as two linear PCR-amplified fragments introducing the N85K mutation, and assembled using homologous recombination into the yeast genome. Fragments were obtained by PCR using pAvA103 and pAvA027 as template and AA76/AA153 and AA80/AA154 as primers (Table S1). Moreover, replacement of pFAS1 with pTEF1-luxO_105 was performed by transforming yeast with pAvA125 and one linear PCR-amplified fragment of pTEF1_luxO-105 with 60 bp homology flanks to the up and downstream sequences of pFAS1. Linear fragment was made by PCR using pAvA109 as template and AA164/AA167 as primers. Transformants were obtained using a heat-shock transformation protocol based on the Gietz and Schiest[74,75]. Transformations were plated on SMG -HIS + NAT, SC -HIS -URA or SMG -HIS -URA + NAT depending on the selection markers used. Colonies were restreaked at least once on selective medium. Correct integration was determined by colony PCR using RedTaq MM. After confirmation of correct integration, plasmids were removed from the yeast by growing on non-selective medium overnight and plating dilutions on YPD to obtain single colonies. Verification of removal of each plasmid was performed by transferring all colonies to filter paper (Whatman, grade 1 round filter paper, 150 mm) and subsequently stamping these colonies on new plates on YPD and on each individual selection (i.e., SC-HIS, SC-URA, SC + NAT). For subsequent transformation rounds, strains still containing the Cas9-plasmid were stocked in addition to the plasmid-free constructed yeast strain.

## Library generation
Error-prone PCR (epPCR) was carried out using the Agilent GeneMorph mutagenesis II kit, with 750 ng as template DNA, aiming for 0-4.5 mutations/kb, according to the manufacturer's instructions. One round of error-prone PCR was carried out using primers AA73/AA77 and pAvA103 as template, followed by a regular PCR with primers AA74/AA78 with the PCR product as template, to amplify the library and to obtain a homology flank on each side of the mutated fragment of 120 bp. Homology arms for integration in X-4, including the promoter sequence or the terminator sequence were amplified using AA757/AA76 and AA79/AA80, respectively, with pAvA103 as template. Yeast libraries were obtained using a heat-shock

transformation protocol based on the Gietz and Schiest[74,75], with the following adaptations: 25 mL of exponentially growing yeast cells ($OD_{600}$ of 2.0) were harvested and used in one transformation. Cells were washed in 0.1 M LiAc prior to the addition of the transformation mixture. For the DNA mixture, equimolar amounts of the three linear fragments were used, adding 800 ng of the largest fragment and 1 µg of the gRNA plasmid per transformation. After resuspending the cells in the transformation mixture, the cells were incubated for 15 minutes at 30 °C, prior to 30-minute heat-shock at 42 °C. The total biomass of 16 transformations (corresponding to 400 mL yeast culture of $OD_{600} = 2$) was pooled together after the recovery step and added to 48 mL of SMG -HIS + clonNAT as one library. By plating for single colonies on SMG -HIS + clonNAT right after the transformation, it was determined that the library contained $2.7 * 10^6$ possible variants. Multiple aliquots of the library were stocked after 2-3 days, when single colonies appeared on the SMG -HIS + clonNAT plates.

### Growth-based OFF and ON selection

After transformation, the yeast library was grown for 2–3 days on SMG -HIS + clonNAT. 1 mL of this culture (or from a cryostock vial) was transferred to fresh 50 mL SMG -HIS + NAT and grown for another 24 h. This culture was diluted 1:50 in 5 mL SMD F-Ac medium and grown for 24 h (OFF-selection) in 50 mL CELLSTAR® CELLREACTOR tubes (Greiner Bio-One). Then, the culture was transferred (1:50) to 5 mL SMD -N + acetamide + ligand and grown for 24 h or till growth was observed (ON-selection), and this step was repeated at least one more time. Each subsequent culture was stocked, analysed by flow cytometer and where indicated, plated for single colonies on YPD or YPD + ligand. Mutant genotypes were determined by Sanger sequencing of *GAL4*_AD-NLS-*luxR*.

### FACS-based selection

After sequential transfers on counter-selective and selective medium as described above, *S. cerevisiae* cells containing the *GAL4*_AD-NLS-*luxR* library were grown in 1 mL of SC + 50 nM C8-HSL for 6 h. The cells were analysed on a Sony fluorescence-assisted cell sorting (FACS) instrument with a blue laser (488 nm) to detect yeGFP fluorescence. 10.000 events were recorded and used to gate the 7% most fluorescent population. Cells were sorted in FITC-A versus FSC-A and collected in 5 mL SC. After recovery overnight, the cells were stored at −70 °C in aliquots by adding 25% (v/v) glycerol and plated for single colonies on YPD.

### Flow cytometry

A 1 mL aliquot of glycerol −70 °C freezer stock was used to inoculate 50 mL of SC medium or a single colony from the screened library was inoculated into 200 µL of SC medium and grown for 16 h. For analysis of ligand-dependent activation, the culture was diluted 1:20 in 200 µL SC medium ± inducer in 96-deep-well culture plates and grown for 6 h at 30 °C. For analysis of ligand-dependent repression, the culture was diluted 1:50 and grown for 24 h prior to being diluted 1:20 in 200 µL SC medium ± inducer in 96-deep-well culture plates and grown for 6 h. Cells were washed and diluted 1:4 in PBS prior to analysis by flow cytometer. Flow cytometry analysis was performed on the NovoCyte Quanteon™ (Agilent). 20.000 events were recorded for each well, with a threshold for event detection at >150,000 FSC-H and a core diameter of 10.1 µm. For yeGFP, excitation was performed with a blue laser (488 nm) and emission detection with a 530/30 nm BP filter (471 V). Subsequent analysis was performed using FlowJo licensed software. FSC-A was plotted against FSC-H to gate for singlet events (Fig. S17). Gated events were used to determine the median fluorescence of the population.

### Fluorescence plate reader cultivation

Growth and fluorescence were analyzed using a BioTek Synergy H1 microplate reader. Inocula were prepared as follows: 1 mL aliquot of glycerol -70 °C freezer stock was grown for 16 h in 50 mL of SC medium (pH 5.6) in baffled 250 mL shake flasks and transferred to 5 mL fresh SC-medium in 50 mL CELLSTAR® CELLREACTOR tubes (Greiner Bio-One) and

exponentially growing cells ($OD_{600} = 5$–6) were used to inoculate the plate at a starting $OD_{660}$ of 0.25-0.3. Black clear-bottom 96-well plates (Greiner Bio-One, catalogue no. 655090) were used, with a total volume of 150 µL per well. $OD_{660}$ and yeGFP fluorescence (588/633, gain 80) were recorded every 20 min and temperature was set to 30 °C with double orbital continuous shaking.

### Structural mapping of mutations

The structures of LuxR and Gal4_AD-NLS-LuxR dimerized and bound to C8-HSL and DNA binding sites were generated using Boltz1. Figures indicating the mutated residues were created using PyMol.

### Statistical and data analysis

Analysis of flow cytometric data was performed using FlowJo licensed software. FSC-A was plotted against FSC-H to gate for singlet events. Gated events were used to determine the median fluorescence of the population. Graphs were plotted and statistical analysis was performed using GraphPad Prism V10.4.0 (GraphPad software). Comparison of the different QS-circuits (Fig. 4) was performed by averaging the fluorescence/OD over the first 2-4 hour time window, to establish a baseline. Subsequent fluorescence/OD measurements were then normalized to this baseline average. A circuit was considered induced when four consecutive measurements showed at least a 1.8-fold increase relative to the baseline. The threshold of 1.8-fold was chosen because it represents the maximum average fold-change observed in the least active circuit (N86K).

### AHL and aloesone analysis by TripleQuad LC-MS/MS

Detection and quantification of acylated homoserine lactones and aloesone were determined from yeast supernatant samples. Samples for AHL quantification were undiluted and samples for aloesone determination were 25-fold diluted with deionized water and subjected to analysis by liquid chromatography coupled to tandem mass spectrometry. Briefly, chromatography was performed on a 1290 Infinity II UHPLC system (Agilent Technologies, Germany). Separation was achieved on a Zorbax Eclipse-Plus C18 column (50 × 3.0, 1.8 µm, Agilent Technologies). Formic acid (0.05%, v/v) in water and acetonitrile (supplied with 0.05% formic acid, v/v) were employed as mobile phases A and B respectively. The elution profile for detection of AHLs and aloesone was: 0–0.3 min, 10% B; 0.3–4.0 min, 10–98% B; 4.0–5.0 min 98% B; 5.0–5.10 min, 98–10% B and 5.1-6.0 min 10% B. The mobile phase flow rate was 400 µL min$^{-1}$. The column temperature was maintained at 40 °C. The liquid chromatography was coupled to an Ultivo Triplequadrupole mass spectrometer (Agilent Technologies) equipped with a Jetstream electrospray ion source (ESI) operated in positive ion mode. The instrument parameters were optimized by infusion experiments with pure standards. The ion spray voltage was set to 3000 V. Dry gas temperature was set to 325 °C and dry gas flow to 10 L min$^{-1}$. Sheath gas temperature was set to 400 °C and sheath gas flow to 12 L min$^{-1}$. Nebulizing gas was set to 45 psi. Nitrogen was used as a dry gas, nebulizing gas and collision gas. Multiple reaction monitoring (MRM) was used to monitor precursor ion → fragment ion transitions. MRM transitions were determined by direct infusion experiments of reference standards. Detailed values for mass transitions can be found in Supplementary Table S6. Both Q1 and Q3 quadrupoles were maintained at unit resolution. Mass Hunter Quantitation Analysis for QQQ software (Version 10, Agilent Technologies) was used for data processing. Linearity in ionization efficiency was verified by analyzing dilution series that were also used for quantification of AHLs in the samples.

### AHL and aloesone analysis by Quadrupole-time-of-flight (Q-TOF) LC-MS/MS

Samples for Q-TOF analysis were prepared similarly to the TripleQuad using the same 25-fold dilution in deionized water. The chromatographic separation was done on a 1290 Infinity II UHPLC system (Agilent Technologies) equipped with a Zorbax Eclipse XDB-C18 column (100 × 3.0 mm, 1.8 µm, Agilent Technologies). Formic acid (0.05%, v/v) in water was used as mobile phase A and acetonitrile (supplied with 0.05% formic acid, v/v) as

mobile phase B. The 15-minute gradient was as follows: 0.0–2.0 min, 3% B; 2.0–11.0 min, 3-75% B; 11.0–12.5 min 75–100% B, 12.5–13.5 min 100% B, 13.5–13.6 min 100–3% B and 13.6–15.0 3% B. The mobile phase flow rate was 400 µl/min. The column temperature was maintained at 30 °C. The liquid chromatography was coupled to a Bruker timsToF Pro mass spectrometer (Bruker, Bremen, Germany) equipped with an electrospray ion source (ESI) operated in positive. The ion spray voltage was maintained at +4200 V, the dry temperature was set to 200 °C, and the dry gas flow was set to 8 L/min. Nitrogen was used as the dry gas, nebulizing gas, and collision gas. The nebulizing gas was set to 2.5 bar and collision energy to 10 eV. MS spectra were acquired in an $m/z$ range from 50 to 1500 amu and MS/MS spectra in a range from 50–1500 amu. Sampling rate was 5 Hz in both ion modes. Na-formate clusters were used for mass calibration. All files were calibrated by postprocessing

## Shakeflask cultivation

Aerobic shakeflasks cultivation was performed in baffled 250 mL shake flasks, with a working volume of 50 mL. 1 mL aliquot of glycerol -70°C freezer stock was used to inoculate 50 mL of SC medium (pH 5.6) in baffled 250 mL shake flasks. After 16 h of growth, this culture was used to prepare the preculture in 50 mL fresh SC-medium. Exponentially growing cells ($OD_{600}$ = 5–6) were used to inoculate shakeflasks at a starting $OD_{660}$ of 0.3–0.4. $OD_{600}$ measurements were performed by sampling from the shakeflasks and measuring on DS-C Cuvette Spectrophotometer (DeNovix).

## BioLector cultivation

pH-controlled fed-batch cultivation was performed in Microfluidic FlowerPlates (m2plabs, Beckman Coulter Life Sciences; Lot nr. 2309221) in the BioLector Pro II system (Beckman Coulter Life Sciences). An exponential feed profile was used of $0.48\ \mu L\ h^{-1} * e^{0.0125 * t}$, triggered after 16–20 h. 2 M KOH was used to ensure the pH would not drop below 5.6. The temperature was set to 30 °C and shaking to 1000 rpm. The relative humidity in the growth chamber was maintained at 85% using distilled water to minimize evaporation of the media. Measurements of biomass and pH were performed every 4 min. A starting volume of 800 µL start medium was added in each well and 1800 µL in each well was designated for feed and base control. Calibration values corresponding to the Lot nr. were used for pH. Precultures were prepared by inoculating 1 mL aliquot of glycerol −70°C freezer stock in 50 mL of SC medium (pH 5.6) in baffled 250 mL shake flasks and cells were grown for 16 h before being diluted in 5 mL fresh SC-medium in 50 mL CELLSTAR® CELLREACTOR tubes (Greiner Bio-One) and exponentially growing cells ($OD_{600}$ = 5–6) were used to inoculate the plate at a starting $OD_{600}$ of 0.25–0.3.

## Statistics and reproducibility

For flow cytometry analysis, 2–3 biological replicates were used and 20,000 events were recorded. Data were normally distributed (Fig. S18) and the obtained results were highly reproducible; i.e., internal controls (control yeast strains and/or control conditions) were always included in each experiment and performed similarly for each independent experiment. For the biolector fedbatch experiment, we used 4–5 biological replicates and 3 technical replicates for LC-MS analysis, to conclude consistent and significant differences between the strains. Significance of aloesone production (Fig. 5) was analysed by one-way ANOVA, with strain type as the independent variable. Post hoc analysis was performed using Tukey's HSD test to evaluate pairwise comparisons between strains.

## Reporting summary

Further information on research design is available in the Nature Portfolio Reporting Summary linked to this article.

## Data availability

All data shown in figures are available in the Source data provided with this paper. There are no restrictions on data availability. Source data behind all figures in the main text are in Supplementary data 1 and Source data for all figures in the Supplementary information are provided in Supplementary data 2. Plasmids pAvA139 (*GAL4*_AD-*luxR*_gen1), pAvA141 (*GAL4*_AD-*luxR*_gen1) and pAvA143 (*GAL4*_AD-*luxR*_gen1) can be obtained from Addgene (Addgene plasmids #248145, #248146 and #248147, respectively). Any other material can be obtained upon request.

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

## Acknowledgements
This study is part of the project *Orthogonal quorum-sensing systems in yeast cell factories* with file number 019.231EN.007 of the Rubicon research programme, which is financed by the Dutch research council (NWO) and awarded to AA. MH is funded by the Novo Nordisk Foundation, grant number NNF22SA0078231 (Copenhagen Bioscience PhD Programme). This project has received funding from the Novo Nordisk Foundation, grant number NNF20CC0035580. We thank Robert Mans for fruitful discussions about the screening workflow. We thank Emma Hoch-Schneider for support during laboratory onboarding. We thank Arsenios Vlassis for technical support on the flow cytometer. We thank Divya Dharshini Uma Shankar and Beata Lehka for technical support on the biolector. We thank Ditte Hededam Welner for kindly supplying us with aloesone standard.

## Author contributions
A.A.: Conceptualization, Funding acquisition, Investigation, Visualization, Formal analysis, Methodology, Project administration, Writing – original draft, Writing – review & editing. M.H.: Formal analysis, Visualization, Writing – review & editing. M.L.J.: Investigation. T.S.: Investigation. C.C.: Resources, Investigation MP: Resources, Investigation. C.A.-R.: Resources. E.D.J.: Resources, Writing – review & editing. M.K.J.: Supervision, Writing – review & editing. All authors reviewed and approved the final version of the manuscript.

## Competing interests
The authors declare no competing interests.
