## [Transparent Peer Review file · Communications Biology]

Engineering N-acyl-homoserine lactone-based quorum-sensing circuit for dynamic regulatory control in *Saccharomyces cerevisiae*

Corresponding Author: Dr Aafke van Aalst

Version 0:

Reviewer comments:

Reviewer #1

(Remarks to the Author)

This study presents an orthogonal quorum sensing (QS) system in *Saccharomyces cerevisiae* using engineered AHL signaling and evolved LuxR variants to enable cell density-dependent gene regulation. The system was applied to improve aloesone production by 51% through QS-mediated repression, demonstrating its potential for dynamic pathway control in yeast metabolic engineering. The study is largely technically sound but suffers from some minor issues. The following comments need to be fully addressed before the paper can be accepted for publication.

Comments:

1. Line 107-109, it might be helpful to also test (or maybe the authors have tested but did not report it) whether *S. cerevisiae* is capable of degrading C6-HSL and C12-HSL. This additional experiment could further support the interpretation that these genes are not functional in yeast cells.
2. The exposure of *S. cerevisiae* to C6-HSL and C12-HSL has been reported to induce morphological and phenotypic changes (<https://link.springer.com/article/10.1186/s13568-016-0292-y>). Since the authors are producing C8-HSL and have observed growth defects when increasing its titer (lines 121–125), it may be worthwhile to examine the morphology of the final strain. This could help clarify whether the reduced cell density is due to the bioactivity of C8-HSL itself or simply a consequence of the metabolic burden from overexpressing SAM2 and MET6.
3. In Figure 1C, error bars are missing for some data categories. Additionally, the use of black error bars on a black bar graph makes them difficult to distinguish. The authors are encouraged to revise the figure to ensure that all data are clearly presented and easily interpretable.
4. In Figure 3C, it appears that many variants exhibiting enhanced dynamic ranges also show increased leaky activity. Additionally, the average leaky activity among variants with wide dynamic ranges seems to rise progressively from T1 to T4. Could the authors provide an explanation for this trend? For instance, might it be related to the inherent requirement for high *amdS* expression in ON-selection conditions (lines 177–180)? This potential limitation could negatively impact the broader applicability of the selection system, and it would be helpful to address this point in the discussion.
5. In Figure 2A, 2B, 4B and 4D, biological triplicates are required to generate error bars for these curves to ensure reproducibility.

Reviewer #2

(Remarks to the Author)

The authors present an effective quorum-sensing system in *Saccharomyces cerevisiae* that functions as an autoinducible dynamic regulation platform. The work encompasses two key developments: a C8-HSL-based sender module created through metabolic engineering and an optimized C8-HSL biosensor enhanced via directed evolution. The biosensor optimization employed an elegant *amdS*-based selection and counter-selection platform, enabling ultra-high throughput screening of millions of biosensor mutants generated by error-prone PCR. This approach yielded a biosensor with over an order of magnitude improvement in sensitivity, reducing the operational range from >500 nM to <10 nM. The authors further demonstrated the system's versatility by showing that the same biosensor can function for both gene activation and repression simply by removing the activation domain. As a proof-of-concept application, they achieved improved aloesone

production by dynamically repressing the fatty acid biosynthesis pathway to enhance flux toward product synthesis. This is a well-executed study with comprehensive analysis and compelling findings. The manuscript merits publication in Communications Biology following consideration of the minor comments below.

The following comments are directed toward the conclusions of the manuscript:

- The authors report up to 20% decreases in growth rate and final cell density upon sender module introduction (lines 122-123). However, Figure 5D suggests that aloesone titers remain unchanged with and without the sender module. Please clarify whether growth impacts observed during sender module characterization also occurred during aloesone production experiments, and if so, discuss why production titers were unaffected.
- While the quorum-sensing system represents an exciting advancement, the growth defects associated with the sender module—whether due to metabolic burden or C8-HSL toxicity—could limit production capacity and cause stability issues during extended fermentations. A discussion of these potential limitations and strategies to mitigate them would strengthen the manuscript and guide future applications.
- The biosensor optimization focused primarily on gene expression activation, yet the applied demonstration utilized only the repression function (without activation domain). Including an additional application experiment showcasing dynamic gene activation for improved bioproduction would better demonstrate the system's full potential and broaden its appeal to the synthetic biology community.

The following comments are directed toward the clarity of the manuscript:

- In line 415, should the unit for aloesone production be μM ? If nM is right, y-axis in Fig. 5D should be corrected.

Reviewer #3

(Remarks to the Author)

The manuscript describes for the first time the full establishment of a bacterial AHL-based quorum sensing system in yeast, its improvement through directed evolution of the transcription factor component and a subsequent application in polyketide production. This is a nice study with high potential for future applications.

1. When the authors tested the initial AHL synthases and did not detect any products, I assume another explanation for this would be the lack of suitable acyl donors (line 106-109)? The authors later discuss the discrepancy between ACP- and CoA-bound precursors. In addition, the 3-oxo-chains are even less likely to be released from fatty acid synthase than the straight C8-chain and intermediates of beta-oxidation will remain in the peroxisomes. When detecting C8-HSL, did the author observe any by-products of different chain-length as well?
2. Did the authors try to use the cognate transcription factor from *B. cenocepacia* instead of the one from *V. fischeri*?
3. Line 232-233: When looking at figure S5, there may still be a slight enrichment in favor of the pGAL_core and against the pTEF construct (due to differences in metabolic burden?). Maybe re-phrase to “no major enrichment”.
4. It would be useful to include (as supplementary material) the corresponding growth curves of the plate reader time-course measurements (Figure 4D). Did the authors conduct a time-course for the repression construct as well?
5. When testing LuxR as repressor, why did the authors remove the activation domain? The full construct would allow for dual use as activator and repressor in the same cell. Like this, it would also have been possible to simultaneously induce the PKS gene in the production strain.
6. Line 389-390: The second operator site did not seem to have increased the dynamic range, but the absolute fluorescence values seem to have changed. This might be worth mentioning.
7. I recommend changing the font in all figures to one that is easier to read (any sans serif type). The currently chosen one becomes problematic specifically when the writing is small. Please also make sure illustration components are explained at least once, e.g. the activation domain in 3B, the metabolites in 4C or the RNA polymerase in 5A.
8. Please carefully check captions of Figure 3 and Figure S7 for both language and clarity. These are currently not easy to understand. Double-check all other figure captions as well, e.g. for starting titles with a capital letter (S1, S2) and language (last two sentences in S6).
9. The discussion is rather short, especially with regard to the potential effect of the selected mutations. The authors may want to expand on this topic. An aspect that is not mentioned or discussed is the fact that the mutant versions also seem to have increased base-line expression of the target gene. Another topic worth discussing would be how to further tune the system and/or how to establish systems dependent on acyl-ACP donors not natively available in yeast.
10. Line 571: Did the authors check the actual mutation rate?

11. Figure S1: It is not clear to me whether the figure shows total ion or extracted chromatograms. The authors should include a negative control and/or the extracted mass spectrum of both standard and supernatant peak.
12. Figure S2B: Strain Cepl is mentioned in the caption, but not included in the figure.
13. Figure S9: Which strain was used as control here?
14. Line 113: Change to "at 7 nM".
15. Line 117 and 119: Delete "driven by".
16. Line 123 and 124: Change "high" to "strong".
17. Line 130: Change "detection" to "addition".
18. Line 177: Change "for growth" to "to grow".
19. Line 181: I suggest changing to "since it is converted by AmdS to the toxic product fluoroacetate".
20. Line 203, 207, 222: When quantifying a level of reduction, please express this in percentage, not fold (as you already do in line 211).
21. Line 221-223: "Single cassette" sounds a bit odd in this context as a single expression cassette is used in either case. Please re-phrase to e.g. "separate protein" or similar.
22. Line 255: Only Med2, but not VP48 seems to have been investigated in the given reference.
23. Line 378: Delete "in yeast" (repetition).
24. Line 386: What do the authors mean with "synergistic effect" here (comparing what exactly)?
25. Check method section for correct gene nomenclature.
26. Line 495: A reference is missing.
27. I would recommend changing "hours" and "minutes" to "h" and "min" throughout the text.

Version 1:

Reviewer comments:

Reviewer #1

(Remarks to the Author)

All of my comments have been fully addressed. I have no further comments. The paper can be accepted for publication.

Reviewer #2

(Remarks to the Author)

All comments were addressed sufficiently.

Reviewer #3

(Remarks to the Author)

The authors addressed all my comments adequately.

Reviewers' comments:

Reviewer #1 (Remarks to the Author):

This study presents an orthogonal quorum sensing (QS) system in *Saccharomyces cerevisiae* using engineered AHL signaling and evolved LuxR variants to enable cell density-dependent gene regulation. The system was applied to improve aloesone production by 51% through QS-mediated repression, demonstrating its potential for dynamic pathway control in yeast metabolic engineering. The study is largely technically sound but suffers from some minor issues. The following comments need to be fully addressed before the paper can be accepted for publication.

We thank the reviewer for the thoughtful feedback and have addressed all comments below.

Comments:

1. Line 107-109, it might be helpful to also test (or maybe the authors have tested but did not report it) whether *S. cerevisiae* is capable of degrading C6-HSL and C12-HSL. This additional experiment could further support the interpretation that these genes are not functional in yeast cells.

*To address the potential degradation of AHL molecules in *S. cerevisiae*, we have now included a time course analysis of C8-HSL concentrations in yeast culture (see Supplementary Figure 2A). This data shows that C8-HSL remains stable over the tested time period during the stationary phase, indicating that *S. cerevisiae* does not actively degrade this molecule. This is now clarified in lines 124-127 of the results section.*

*AHL molecules such as C6, C8, and C12-HSL differ only by their acyl chain length, and yeast is not known to possess AHL-lactonase or AHL-acylase activity and especially not some that would discriminate so selectively. Therefore, we believe it is reasonable to assume that C6-HSL and C12-HSL are also not subject to degradation in *S. cerevisiae* under the conditions tested.*

To further support this, we also constructed a strain carrying a fast-turnover GFP reporter in combination with Gal4_AD-LuxR_gen1 and monitored GFP stability following addition of C6-HSL. The GFP signal remained stable over time, consistent with the absence of degradation. Because this assay is more indirect and relies on components introduced later in the manuscript (ie. the biosensor), we did not include the figure in the final version, but we note it here to reassure the reviewer that AHL degradation is not occurring.

2. The exposure of *S. cerevisiae* to C6-HSL and C12-HSL has been reported to induce morphological and phenotypic changes (<https://link.springer.com/article/10.1186/s13568-016-0292-y>). Since the authors are producing C8-HSL and have observed growth defects when increasing its titer (lines 121–125), it may be worthwhile to examine the morphology of the final strain. This could help clarify whether the reduced cell density is due to the bioactivity of C8-HSL itself or simply a consequence of the metabolic burden from overexpressing SAM2 and MET6.

*Upon examining the cited study, we note that while morphological changes were observed in *S. cerevisiae* exposed to 2 μM of C6-HSL or C12-HSL, no significant effects on growth rate or final cell density were reported, even at this relatively high concentration. Moreover, at 200 nM, no morphological or growth-related changes were observed. Since the concentrations of C8-HSL produced in our system are well below 2 μM (and 200 nM), it seems unlikely that the observed growth phenotype is a consequence of C8-HSL bioactivity. To further confirm this, we cultured yeast in the presence and absence of AHLs (100 μM) and we did not see any changes in growth rate or final OD, which you can now find in Fig.S2C. We have clarified this point in lines 116-119 of the manuscript in which we now include the suggested article as well.*

3. In Figure 1C, error bars are missing for some data categories. Additionally, the use of black error bars on a black bar graph makes them difficult to distinguish. The authors are encouraged to revise the figure to ensure that all data are clearly presented and easily interpretable.

We have revised the figures to make the bar graph more readable. Error bars were already included, but seemed missing for some of the graphs, since the replicates yielded the exact same concentration of C8-HSL.

4. In Figure 3C, it appears that many variants exhibiting enhanced dynamic ranges also show increased leaky activity. Additionally, the average leaky activity among variants with wide dynamic ranges seems to rise progressively from T1 to T4. Could the authors provide an explanation for this trend? For instance, might it be related to the inherent requirement for high *amdS* expression in ON-selection conditions (lines 177–180)? This potential limitation could negatively impact the broader applicability of the selection system, and it would be helpful to address this point in the discussion.

Indeed, we observed that several variants with increased dynamic range also displayed elevated background activity. However, we do not believe this is a direct artifact of the selection system. Notably, a substantial fraction of variants with consistently low expression (i.e., always-OFF biosensors) remained present in the selected pools, even after ON-selection. This suggests that biosensors with low basal activity are not necessarily outcompeted during ON-selection, particularly in the early rounds.

It is plausible that during multiple consecutive transfers, variants with the highest induced expression levels—regardless of their OFF-state—might become enriched. However, we did not observe mutants that exhibit an always ON phenotype, and no new genetic variants emerged in later transfers. Thus, the average leaky activity across variants did not progressively increase due to selection pressure, but the abundance of variants with increased dynamic range did increase.

The observed trend may instead reflect an inherent trade-off between lowering activation threshold and maintaining tight OFF-state. This balance is a common challenge in the design of responsive transcription factor-based biosensors. We have now addressed this point in the discussion (lines 473–476).

5. In Figure 2A, 2B, 4B and 4D, biological triplicates are required to generate error bars for these curves to ensure reproducibility.

For figure 2A, 2B and 4D (and all other figures with growth curves), we have now represented the data as average with error bars (instead of one representative replicate and the other replicates in the supplementary). For figure 4B, we have repeated the experiment and the graph is now showing data from n=4, as indicated in the figure description.

Reviewer #2 (Remarks to the Author):

The authors present an effective quorum-sensing system in *Saccharomyces cerevisiae* that functions as an autoinducible dynamic regulation platform. The work encompasses two key developments: a C8-HSL-based sender module created through metabolic engineering and an optimized C8-HSL biosensor enhanced via directed evolution. The biosensor optimization employed an elegant amdS-based selection and counter-selection platform, enabling ultra-high throughput screening of millions of biosensor mutants generated by error-prone PCR. This approach yielded a biosensor with over an order of magnitude improvement in sensitivity, reducing the operational range from >500 nM to <10 nM. The authors further demonstrated the system's versatility by showing that the same biosensor can function for both gene activation and repression simply by removing the activation domain. As a proof-of-concept application, they achieved improved aloesone production by dynamically repressing the fatty acid biosynthesis pathway to enhance flux toward product synthesis.

This is a well-executed study with comprehensive analysis and compelling findings. The manuscript merits publication in *Communications Biology* following consideration of the minor comments below.

We thank the reviewer for the thoughtful words and the kind remarks. We have incorporated all the feedback aimed for the conclusion of the manuscript.

The following comments are directed toward the conclusions of the manuscript:

- The authors report up to 20% decreases in growth rate and final cell density upon sender module introduction (lines 122-123). However, Figure 5D suggests that aloesone titers remain unchanged with and without the sender module. Please clarify whether growth impacts observed during sender module characterization also occurred during aloesone production experiments, and if so, discuss why production titers were unaffected.

We have addressed this comment by including the growth profiles during the aloesone experiment in fedbatch cultivation in the supplementary materials and we now mention in the text that growth impacts were not observed (lines 418-419) and included the growth profiles in Figure S13.

- While the quorum-sensing system represents an exciting advancement, the growth defects associated with the sender module—whether due to metabolic burden or C8-HSL toxicity—could limit production capacity and cause stability issues during

extended fermentations. A discussion of these potential limitations and strategies to mitigate them would strengthen the manuscript and guide future applications.

We have now addressed these limitations and discussed potential mitigation strategies in the discussion (lines 448-453 and lines 462-465).

- The biosensor optimization focused primarily on gene expression activation, yet the applied demonstration utilized only the repression function (without activation domain). Including an additional application experiment showcasing dynamic gene activation for improved bioproduction would better demonstrate the system's full potential and broaden its appeal to the synthetic biology community.

We agree that extending the activator to a production context is a promising future direction and now mention this in the discussion (lines 479-487).

The following comments are directed toward the clarity of the manuscript:

- In line 415, should the unit for aloesone production be μM ? If nM is right, y-axis in Fig. 5D should be corrected.

I have changed the graph to have aloesone production in μM on the left y-axis, and the C8-HSL production on the right y-axis.

Reviewer #3 (Remarks to the Author):

The manuscript describes for the first time the full establishment of a bacterial AHL-based quorum sensing system in yeast, its improvement through directed evolution of the transcription factor component and a subsequent application in polyketide production. This is a nice study with high potential for future applications.

We thank the reviewer for the thoughtful feedback and have addressed all comments below and implemented all suggested changes in the manuscript and supplementary materials.

1. When the authors tested the initial AHL synthases and did not detect any products, I assume another explanation for this would be the lack of suitable acyl donors (line 106-109)? The authors later discuss the discrepancy between ACP- and CoA-bound precursors. In addition, the 3-oxo-chains are even less likely to be released from fatty acid synthase than the straight C8-chain and intermediates of beta-oxidation will remain in the peroxisomes. When detecting C8-HSL, did the author observe any by-products of different chain-length as well?

Indeed, we agree with the reviewer that an important limitation is the availability of acyl-donors. We did not observe any by-products of different chain lengths.

2. Did the authors try to use the cognate transcription factor from *B. cenocepacia* instead of the one from *V. fischeri*?

Yes, in addition to testing multiple synthases, we have aimed to engineer multiple biosensors using different transcription factors. CepR2 from Bc has been introduced as well, but was not found to be responsive. However, often for functional implementation of prokaryotic TFs additional engineering is required.

3. Line 232-233: When looking at figure S5, there may still be a slight enrichment in favor of the pGAL_core and against the pTEF construct (due to differences in metabolic burden?). Maybe re-phase to “no major enrichment”.

We agree and have implemented the suggested change.

4. It would be useful to include (as supplementary material) the corresponding growth curves of the plate reader time-course measurements (Figure 4D). Did the authors conduct a time-course for the repression construct as well?

We have included the corresponding growth curves as supplementary material. The QS-controlled repression was only tested in the aloesone testbed.

5. When testing LuxR as repressor, why did the authors remove the activation domain? The full construct would allow for dual use as activator and repressor in the same cell. Like this, it would also have been possible to simultaneously induce the PKS gene in the production strain.

We thank the reviewer for the insightful suggestion to retain an activation domain in LuxR to enable potential dual-use as both activator and repressor. In early exploratory experiments, we tested a LuxR fusion with the VP48 activation domain in our promoter/operator configuration. While LuxR without an activation domain produced robust repression, the VP48–LuxR fusion did not retain repression and instead activated transcription, making it unsuitable for the repressor role in our study. For this reason, we proceeded with the repressor-only LuxR construct in all downstream experiments to ensure consistent repression of target genes. We have now added the results as well as the reasoning in the manuscript in lines 143-161.

We agree that developing a dual-purpose LuxR variant, for example by testing alternative activation domains such as GAL4–AD, could be valuable in future work, especially for multiplexed regulation in a single strain. We have also added a brief note in the discussion (479-487) to reflect this potential future avenue.

6. Line 389-390: The second operator site did not seem to have increased the dynamic range, but the absolute fluorescence values seem to have changed. This might be worth mentioning.

We have implemented this suggestion and clarified this in lines 400-401.

7. I recommend changing the font in all figures to one that is easier to read (any sans serif type). The currently chosen one becomes problematic specifically when the writing is small.

Please also make sure illustration components are explained at least once, e.g. the activation domain in 3B, the metabolites in 4C or the RNA polymerase in 5A.

We have adjusted all figures for clear font types. We have included explanation to the illustration components as well.

8. Please carefully check captions of Figure 3 and Figure S7 for both language and clarity. These are currently not easy to understand. Double-check all other figure captions as well, e.g. for starting titles with a capital letter (S1, S2) and language (last two sentences in S6).

We have revised all captions.

9. The discussion is rather short, especially with regard to the potential effect of the selected mutations. The authors may want to expand on this topic. An aspect that is not mentioned or discussed is the fact that the mutant versions also seem to have increased base-line expression of the target gene. Another topic worth discussing would be how to further tune the system and/or how to establish systems dependent on acyl-ACP donors not natively available in yeast.

We have expanded on the discussion by including thoughts on further improving availability of acyl-donors (lines 458-460). Moreover, we have further touched upon the base-line expression caused by the mutations (lines 473-476).

10. Line 571: Did the authors check the actual mutation rate?

We did not check the mutation rate, but we were using a commercial kit for epPCR and used the settings to aim for 2 mutations.

11. Figure S1: It is not clear to me whether the figure shows total ion or extracted chromatograms. The authors should include a negative control and/or the extracted mass spectrum of both standard and supernatant peak.

We have included the suggested mass spectra of the negative control (non-engineered yeast supernatant) in the figure.

12. Figure S2B: Strain Cep1 is mentioned in the caption, but not included in the figure.

13. Figure S9: Which strain was used as control here?

14. Line 113: Change to "at 7 nM".

15. Line 117 and 119: Delete "driven by".

16. Line 123 and 124: Change "high" to "strong".

17. Line 130: Change "detection" to "addition".

18. Line 177: Change "for growth" to "to grow".

19. Line 181: I suggest changing to "since it is converted by AmdS to the toxic product fluoroacetate".

20. Line 203, 207, 222: When quantifying a level of reduction, please express this in percentage, not fold (as you already do in line 211).

21. Line 221-223: "Single cassette" sounds a bit odd in this context as a single expression cassette is used in either case. Please re-phrase to e.g. "separate protein" or similar.

We have implemented all suggested changes.

22. Line 255: Only Med2, but not VP48 seems to have been investigated in the given reference.

We have checked the reference and indeed, it is VP64 they have tested, not VP48. We have changed the text accordingly.

23. Line 378: Delete “in yeast” (repetition).

24. Line 386: What do the authors mean with “synergistic effect” here (comparing what exactly)?

25. Check method section for correct gene nomenclature.

26. Line 495: A reference is missing.

27. I would recommend changing “hours” and “minutes” to “h” and “min” throughout the text.

We have implemented and clarified all comments in the main text.